# CAUSAL-TGAN: MODELING TABULAR DATA USING CAUSALLY-AWARE GAN

**Bingyang Wen, Yupeng Cao, Fan Yang, K.P. Subbalakshmi & R. Chandramouli**
Department of Electrical and Computer Engineering
Stevens Institute of Technology
Hoboken, NJ 07030, USA
`{bwen4, ycao33, fyang14, ksubbala, mouli}@stevens.edu`

## ABSTRACT

Generative adversarial net (GAN)-based tabular data generation has recently received significant attention for its power for data augmentation when available data is limited. Most prior works have applied generic GAN frameworks for tabular data generation without explicitly considering inter-variable relationships, which is important for modeling tabular data distribution. In this work, we design Causal-TGAN, a causally-aware generator architecture that can capture the relationships among variables (continuous-type, discrete-type, and mixed-type) by explicitly modeling the pre-defined inter-variable causal relationships. The flexibility of Causal-TGAN is its capability to support different degrees of subject matter expert domain knowledge (e.g., complete or partial) about the inter-variable causal relations. Extensive experimental results on both simulated and real-world datasets demonstrate that exploiting causal relations in deep generative models could improve the generated tabular data quality compared to the state-of-the-art. Code is available at https://github.com/BiggyBing/Causal-TGAN-Public.

## 1 INTRODUCTION

Tabular data is one of the most common types of data, and it is used in a wide range of applications, including medical diagnosis (Ulmer et al., 2020), financial applications (e.g., risk analysis and investment strategy recommendation) (Clements et al., 2020), fraud detection (Cartella et al., 2021), recommendation systems (Sun et al., 2019) etc. However, the availability of high-quality tabular data is sometimes limited due to incomplete records or the high cost associated with data collection. As a remedy, tabular data synthesis has recently received significant attention.

Prior works of GANs for image/language demonstrate that designing a specialized network that can capture the correlations among features has significant advantages (e.g., (Radford et al., 2015; Yu et al., 2017)). But, note that correlation among features in tabular data could potentially be weaker than spatial (for images) or semantic (for language data) relationships. This is because images or languages contain position-related correlations that may not hold for tabular data. However, unlike image or language data, tabular datasets are usually well-structured, and the values of each column (feature) in the tabular data are usually measurements with a physical meaning, such as age or income. This motivates us to ask if the interaction between the features can be captured in some way and used in the generation process.

We believe that the causal relationships between these variables can be the answer to the above question. Causality among the features of a table can indicate how these features interact with each other and then progress to produce the tabular data. More formally, the common cause principle (Spirtes et al., 2000b) states that every correlation is either due to a direct causal effect linking the correlated entities or is brought about by a third factor, a so-called common cause. For example, there could be three possible causal relations between variables $A$ and $B$ to make them dependent, i.e., $A$ causes $B$, $B$ causes $A$, or another variable $C$ causes both $A$ and $B$. It is now clear that the interaction between the features can be captured by causality. The interactions among a set of random variables, to some extent, entail the joint distribution of these variables.

In this work, we show that exploiting these causal relations in deep generative models delivers synthesized data that more accurately captures the target data distribution. Additionally, we use special treatments to deal with common issues in tabular data generation (see Section 4.1). We name our method **Causal-TGAN** (Causally-aware Tabular Data Generative Neural Net). Causal-TGAN explicitly models inter-feature causal relationships (usually described via a causal graph ) via a causally-aware generator. We can treat the true causal graph as expert knowledge. The expert knowledge can be either obtained from domain experts or data by using the causal graph discovery methods (Glymour et al., 2019). The availability of accurate and complete knowledge (i.e., causal relations known for all variables) sometimes can be difficult. Therefore, we propose a hybrid generative mechanism for data generation when causal relations are known only for some variables (Section 4.2).

We summarize our contributions as follows: (i) Causal-TGAN, a tailored architecture for tabular data generation, that can incorporate (complete and incomplete) inter-variable causal relationships from a domain expert; (ii) Detailed experimental results demonstrating that Causal-TGAN is better (compared to other methods) at capturing the target data distribution on both simulated and real-world datasets.

## 2 RELATED WORK

**GANs for Tabular Data Generation**    Prior work has exploited various methods for improving GAN-based tabular data generation. The improvements are made by either prepossessing the variable to make its distribution easier to be modeled by GANs (Xu et al., 2019), augmenting the GAN frameworks (Park et al., 2018; Kim et al., 2021), or both (Zhao et al., 2021). The study that considers inter-variable correlations includes CorGAN (Torfi et al., 2020), which leverages the one-dimensional convolutional GAN architecture to capture the correlation in electronic healthcare records (EHR). However, EHR data have a stronger positional correlation, which a patient's health condition is correlated to the past. It remains unclear how CorGAN will perform on data without such positional correlation, which is common in tabular data. Our Causal-TGAN explicitly encodes the inter-variable causal correlations and is considered more general than positional relations.

**Causally-aware Generator**    Besides Causal-TGAN, we explicitly leave out CGNN (Goudet et al., 2018), CausalGAN (Kocaoglu et al., 2017) and DECAF (van Breugel et al., 2021), which incorporate similar causally-aware generator. Though similarity in generator structure, their objectives are different and hence the applications of the causality-driven generators are explored in different aspects. CGNN's goal is to infer the causal structures; Causal-GAN and DECAF generate interventional data which do not exists in the original data distribution but are causally reasonable. In contrast, our Casual-TGAN's goal is to better model the in-distribution (i.e., original data distribution) by taking advantage of inter-variable causal relations. In addition to the causality-driven generator architecture, we consider to use variable encoding for dealing with irregular variable distribution and a hybrid generation mechanism to facilitate its practicality.

## 3 BACKGROUND

**Structural Causal Models**    Mathematically, an structural causal models (SCM) Pearl (2009) $\mathcal{M}_{\mathcal{G}}$ with a causal graph $\mathcal{G}$ can be represented by a triplet $\mathcal{M}_{\mathcal{G}} = \langle \mathcal{X}, \mathcal{F}, \mathcal{U} \rangle$ that contains a set of endogenous variables $\mathcal{X} = \{X_1, X_2, ..., X_d\}$, a set of causal equations (mechanisms) $\mathcal{F} = \{f_1, f_2, ..., f_d\}$ and a set of exogenous variables $U = \{U_1, U_2, ..., U_d\}$, where each $U_i$ is independently sampled from a distribution $\mathcal{U}$. The causal relationship "$X_i$ causes $X_j$" is represented in the causal graph by a directed edge that orientates from $X_i$ to $X_j$, i.e., $X_i \rightarrow X_j$. The value of $X_j$ is determined by its causal equation $X_j = f_j(Pa_{\mathcal{G}}(X_j), U_j)$ where $Pa_{\mathcal{G}}(X_j)$

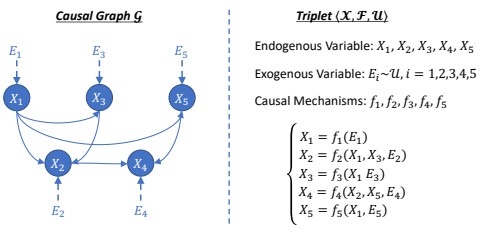

Figure 1: **An example of SCM.** Left: the causal graph, right: the triplet that contains endogenous variables, exogenous variables and causal mechanisms.

denotes all the parent nodes of $X_j$ in $\mathcal{G}$. $U_j$ is
the exogenous variable of $X_j$ and can be seen as the cumulative effect of all unobserved causes of $X_j$. Figure 1 illustrates an example of SCM with 5 variables.

# 4 METHOD

## 4.1 VARIABLE ENCODING

Real-world tabular data present several challenges to tabular data generation, such as mixed-type variables, irregular distribution (e.g., non-Gaussian distribution), and multimodality. The non-Gaussian distribution of continuous variables can cause gradient varnish issues when they are min-max normalized; Moreover, the mode collapse issues of conventional image GANs will also happen to tabular data generation where the continuous variables sometimes have multi-mode. To overcome the problems raised by the complex distribution of a continuous variable, Causal-TGAN embraces the idea of mode-specific normalization (Xu et al., 2019) to continuous variables.

The mode-specific normalization first fits a variational Gaussian mixture model (VGM) for the columns containing continuous variables. Let us assume that the fitted VGM of a column consists of $n$ Gaussian components. Then, a single value from this column can be encoded as a vector of length $n + 1$. The first $n$ elements denote a one-hot vector indicating the most likely Gaussian component that the value belongs to. The last (i.e., $n + 1$st) element is the mean and variance-normalized value of the corresponding Gaussian component. For discrete variables, we use one-hot encoding to encode them. We point out that using variable encoding is one distinct of Causal-TGAN compared with other causality-driven generators.

## 4.2 GENERATOR CONSTRUCTION

**Generator with Causal Graph $\mathcal{G}$**  Causal-TGAN use multilayer perceptron (MLP) to explicitly model an SCM with causal graph $\mathcal{G}$. Each causal mechanism $f_i$ is constructed by a neural network $G_i: \mathbb{R}^{|E(Pa_{\mathcal{G}}(X_i))|+1} \to \mathbb{R}^{|E(X_i)|}$, where $E: \mathbb{R} \to \mathbb{R}^n$ is process of the variable encoding operation that maps a scalar (resp. a set of scalars) into encoded vector (resp. a set of vectors) of length $n$. Values of each variable $X_i$ can be generated by:

$$x_i = G_i(u_i, pa_{\mathcal{G}}(X_i)) \; \forall i \tag{1}$$

where $pa_{\mathcal{G}}(X_i)$ are the generated values of the parent nodes of $X_i$ and $u_i$ is sampled from the pre-defined exogenous variable distribution (e.g. Gaussian) of $X_i$. Following the topological order, one sample can be generated autoregressively as $G(u) = [G_1(u_1, pa_{\mathcal{G}}(X_1)), ..., G_k(u_k, pa_{\mathcal{G}}(X_k))]$.

**Causal-TGAN with Partial Knowledge**  Causal-TGAN can consume only partial knowledge (i.e., causal diagram known only partially for a few variables) for data generation when complete knowledge is not accessible, or domain experts only have high confidence only in partial knowledge. To do this, we first fit Causal-TGAN on a subset of the target dataset containing only the variables with known causal relations. Then we leverage a conditional GAN[1], $C_{cond}$, to generate the rest of the variables conditioned on the variables used by Causal-TGAN:

$$x_S^{cond} = C_{cond}(e, x_S^{causal}) \tag{2}$$

where $e$ is the noise vector. $x_S^{cond}$ and $x_S^{causal}$ are the generated samples from the $C_{cond}$ and $G$ respectively. Then, under the partial knowledge setting a complete data sample is generated by concatenating $x_S^{cond}$ and $x_S^{causal}$. Unlike other SCM-based generators, our new hybrid generating mechanism provides more flexibility in incorporating causal information into SCM-based generators, which facilitates practicality.

## 4.3 TRAINING CAUSAL-TGAN

Either the Causal-TGAN or the Conditional GAN is trained using WGAN loss with gradient penalty (Gulrajani et al., 2017). Note that, when training conditional GAN, we freeze the parameters of the causal-driven generator (i.e., $G$) and update the parameters of the conditional GAN (i.e., $C_{cond}$) only.

---

[1]We use the conditional GAN implementation at https://github.com/sdv-dev/CTGAN

## 5 EXPERIMENTAL RESULTS

### 5.1 SETTINGS

**Datasets**    We use both simulated (causal graph is known) and real datasets (causal graph is unknown) in our experiment. For the simulated dataset, we pick 8 well-known Bayesian Networks to create the simulated datasets of discrete-type, continuous-type, and mixed-type and use their network structures as the true causal graphs. For real datasets, we select 6 mixed-type datasets that are commonly used for classification and regression in machine learning. We use PC algorithms (Spirtes et al., 2000a) to estimate the causal structures for real-world datasets. We summarize the details and statistics of these datasets in Appendix A.

**Baselines**    Several generative models proposed for tabular data—MedGAN (Choi et al., 2017), TableGAN (Park et al., 2018), CTGAN and TVAE (Xu et al., 2019)–were used as the baseline. We use Identity to denote the model that generates training data. Therefore, the method that has a closer performance to Identity is considered better. For all the baseline models, we use the recommended hyperparameters presented in the original papers or provided in their implementations. We train the Causal-TGAN model with a batch size of 500 for 400 epochs. For clarity of presentation, we highlight the best performance in bold font and underline the second-best performance.

**Evaluation Metrics**    For experiments on real-world data, we evaluate the **machine learning efficacy (MLe)** of the generated data. MLe measures how well the synthetic data can be the proxy of the target data in the machine learning tasks. It is measured as the performance on original test datasets of machine learning models which are trained on the generated data. Machine learning models and their settings used for calculating MLe are described in Appendix B. Different than real-world datasets, simulated datasets do not contain machine learning tasks (i.e., labels), which makes it difficult to measure MLe for simulated datasets. Instead, we leverage **Kullback–Leibler divergence (KLD)** and **Log-cluster (LC)** (Goncalves et al., 2020) as metrics that used for simulated datasets. KLD quantifies how much one probability distribution differs from another probability distribution. LC score is intended to evaluate synthetic datasets in unsupervised machine learning tasks. To measure log-cluster, we first concatenate the synthetic dataset and the real dataset into a single dataset. Secondly, a clustering method is applied to the concatenated dataset with a fixed number of clusters $G$. Then the log-cluster score can be calculated as:

$$LC(X_S, X_R) = \log(\frac{1}{G} \sum_{i=1}^{G} [\frac{n_i^r}{n_i} - c]^2) \tag{3}$$

where $n_i$ is the number of samples in the $i^{th}$ cluster and $n_i^r$ is the number of samples in $n_i$ that form the target dataset. $c$ is defined the as ratio of the number of samples in the target dataset to the number of samples in the concatenated dataset. A large value of LC score indicates a severe mismatch in cluster members, indicating a disparity between the distribution of target and synthetic datasets. We set $G$ equal to 100 in our experiments.

### 5.2 RESULTS

The results on both simulated and real-world datasets are reported in Table 1.

**Results on Simulated Datasets**    Notice from the table that Causal-TGAN outperforms all the baseline models on all types of datasets on average. Surprisingly, TableGAN outperforms CTGAN on continuous datasets by a large margin even though TableGAN has no special treatment for continuous variables. One possible reason for this is that the continuous variables in the simulated datasets do not conform to a mixture Gaussian distribution. CTGAN's mixture Gaussian modeling could be introducing extra complexity on variable distribution, which can undermine CTGAN's efficiency. However, note that, even though Causal-TGAN employs the same encoding strategy as CTGAN, it still outperforms TableGAN by incorporating causal knowledge.

**Results on Real-world Datasets**    We train Causal-TGAN with partial knowledge setting to evaluate it on real-world datasets. Specifically, we estimated the causal relations for variables of the majority data type for each dataset. We then use the estimated causal relations for generating the variables of

| Method | Simulated Datasets | | | | | | Real-world Datasets | |
|---|---|---|---|---|---|---|---|---|
| | Discrete | | Continuous | | Mixed | | Mixed | |
| | KLD $\downarrow$ | LC $\uparrow$ | KLD $\downarrow$ | LC $\uparrow$ | KLD $\downarrow$ | LC $\uparrow$ | F1 | $R^2$ |
| Identity | 0 | $\infty$ | 0 | $\infty$ | 0 | $\infty$ | 0.686 | 0.342 |
| MedGAN | 0.406 | 2.311 | 7.088 | 1.407 | 0.901 | 1.784 | 0.230 | 2.970 |
| TableGAN | 0.280 | 3.634 | 0.999 | 6.501 | 0.478 | 2.391 | 0.428 | 0.130 |
| TVAE | 0.104 | 4.276 | 1.635 | 3.846 | 0.098 | 2.834 | 0.460 | 0.238 |
| CTGAN | 0.266 | 3.231 | 1.576 | 2.729 | 0.355 | 2.858 | 0.575 | 0.384 |
| Causal-TGAN | **0.054** | **5.499** | **0.391** | **6.614** | **0.059** | **2.931** | **0.629** | **0.294** |

Table 1: Results on simulated and real-world datasets. For simulated datasets, we report KLD and LC (log-cluster) scores. For real-world datasets, we report F1 scores (for classification datasets) and $R^2$ (for regression datasets) as the MLe scores. The scores are calculated as the average value over all datasets of the same type. Results for each single dataset are reported in Appendix C.

the majority data type and use the conditional generator to generate the rest. The reason for this is that the existing causal discovery methods for mixed-type data either cannot produce causal structures of high accuracy or have an exponential time complexity with respect to the size of the dataset. Our strategy can ensure as much and accurate as knowledge can be delivered into the modeling process. The results on real-world datasets are reported in Table 5. It illustrates that, even when full knowledge is absent, Causal-TGAN outperforms all the baseline models.

## 5.3 ABLATION STUDY

To understand the efficiency of each component in Causal-TGAN, we implement an ablation study on real-world datasets. We try to understand Causal-TGAN in two aspects where there is (1) no special treatment for variables and (2) no expert knowledge. Specifically, when studying Causal-TGAN operating without special treatment for variables, we consider the following settings. (1) w/o M.: continuous variables are standardized into $[-1, 1]$ instead of using mode-specific normalization. (2) w/o G.: use softmax activation function instead of gumbel-softmax function (Jang et al., 2016) for all activation functions that used for generating one-hot vectors. (3) w/o E.: we remove all variable encoding strategies. Instead, we standardize continuous variables and convert categorical variables into integers. Table 2 illustrates the results of ablation study.

| Settings | w/o M. | w/o G. | w/o E. | w/o knowledge |
|---|---|---|---|---|
| Performance | -23.98% | -22.16% | -75.97% | -16.17% |

Table 2: Ablation study results on special treatments on variables and information of inter-variable causal relation. The performance (measured as MLe) changes on real-world datasets are reported.

When there is no expert knowledge in generating real-world datasets, the performance decreases. This result demonstrates the efficiency by incorporating inter-variable causal information. Additionally, in the absence of special treatments for tabular data variables, the best-case, and worst-case performance drop is 21.30% and 75.97%, respectively. Note that removing treatments for variables reduces Causal-TGAN to other causally-aware generators discussed in Section 2. This result illustrates the superior of Causal-TGAN on tabular data generation over other causally-aware generators.

## 6 CONCLUSION

We propose Causal-TGAN, a tabular data generative model that leverages inter-variable causal relationships. This method can handle discrete, continuous, and mixed data types. When combined with an auxiliary conditional GAN, the proposed approach can flexibly consume different types or qualities (complete or partial) of expert knowledge about the underlying causal structures. Extensive experimental evaluation on simulated and real-life datasets indicates superior performance and practicality of Causal-TGAN when compared to several other baseline generative models available in the literature.

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

# A    DATASETS DESCRIPTION

We investigate our model architecture on various simulated and real tabular datasets. In this section, we introduce these datasets and we record data processing steps for some real datasets.

| Dataset Name | # train/test | # Cols. | # B | # C | # R | Dataset Name | # train/test | # Cols. | # B | # C | # R | Task |
|---|---|---|---|---|---|---|---|---|---|---|---|---|
| | | Simulated Dataset | | | | | | Real Dataset | | | | |
| asia | 10k/10k | 8 | 0 | 8 | 0 | adult | 23k/10k | 15 | 2 | 7 | 6 | **C** |
| child | 10k/10k | 20 | 0 | 20 | 0 | census | 200k/100k | 41 | 3 | 31 | 7 | **C** |
| insurance | 10k/10k | 27 | 7 | 20 | 0 | cabs | 37k/5k | 13 | 1 | 5 | 7 | **C** |
| alarm | 10k/10k | 37 | 10 | 27 | 0 | loan | 31k/5k | 17 | 1 | 7 | 9 | **C** |
| ecoli70 | 15k/15k | 46 | 0 | 0 | 46 | news | 31k/8k | 59 | 14 | 0 | 45 | **R** |
| arth150 | 25k/25k | 107 | 0 | 0 | 107 | kings | 16k/5k | 16 | 0 | 2 | 14 | **R** |
| healthcare | 15k/15k | 7 | 0 | 3 | 4 | | | | | | | |
| mehra | 25k/25k | 24 | 0 | 4 | 20 | | | | | | | |

Table 3: Summary of datasets. Note that, # denotes the number of and Col. is the abbreviation of columns. The machine learning tasks for real-world datasets are listed in the last column, where C stands for classification and R stands for regression. We separate the train and test datasets by randomly sampling from the whole dataset.

**Simulated Datasets:** We pick 8 well-known Bayesian Networks as simulated datasets - alarm, asia, child insurance, arth150, ecoli70, healthcare and mehra. Specifically, alarm, asia, child and insurance are generated by Bayesian Networks. Arth150 and ecoli70 are generated by Gaussian Bayesian Networks and hence are of continuous variable type. Healthcare and mehra are generated by Conditional Linear Gaussian Bayesian Networks which contain both discrete and continuous variables.

**Real Datasets:** We select 6 commonly used machine learning datasets. Adult consists of demographic information in the U.S. and is collected by the 1994 Census Survey, where we predict two classes of high (>\$50K) and low ($\leq$\$50K) income. Census is similar to Adult, but it has different columns. Loan is about bank loan status prediction. Cabs is collected by an Indian cab aggregator service company for predicting the types of customers. We select King from Kaggle. Kings dataset contains house sale prices for King County in Seattle between May 2014 and May 2015. News dataset has a heterogeneous set of features about articles published by Mashable in a period of two years for predicting the number of shares in social networks. Adult and census are for binary classification, and cabs are for multi-class classification. Kings and news are for regression.

We process cabs and kings datasets. For cabs, we only use data from train datasets since only train datasets contains labels. For kings datasets, we remove irrelevant columns. The details of data processing are as follow:

- **Cabs:** We first only take the training set part of the original dataset. Next, we delete the rows that contain null values and 'Trip_ID' column. Then, we split the processed dataset into train set and test set according to 10% and 90%.

- **King:** We removed 5 irrelevant columns: 'id', 'date', 'zipcpde', 'lat' and 'long'. After that, we split whole dataset into train set and test set according to 20% and 80%.

The statistics of datasets are summarized in Table 3 and the websites for all the datasets are listed as follows:

- Simulated Datasets: http://www.bnlearn.com/bnrepository/

- Adult: http://archive.ics.uci.edu/ml/datasets/adult

- Census: https://archive.ics.uci.edu/ml/datasets/census+income

- Cabs: https://www.kaggle.com/arashnic/taxi-pricing-with-mobility-analytics

- Loan: https://www.kaggle.com/zaurbegiev/my-dataset

- Kings: https://www.kaggle.com/harlfoxem/housesalesprediction

- News: https://archive.ics.uci.edu/ml/datasets/online+news+popularity

## B    SETTINGS OF MACHINE LEARNING MODELS

| Tasks | Model | Description |
|---|---|---|
| Classification | Adaboost | **n_estimators=50**, and others=defaulted values. |
| | Decision Tree | **max_depth=20**, **max_leaf_nodes=50**, and others=defaulted values. |
| | MLP | **hidden_layer_sizes=50**, **early_stopping=True**, and others=defaulted values. |
| Regression | Linear Regression | All settings with defaulted values. |
| | MLP | **hidden_layer_sizes=100**, **early_stopping=True**, and others=defaulted values. |

Table 4: Classifier and Regressor used in the evaluation of Machine Learning Efficacy. The names of all parameters that used in the description are consistent with those defined in scikit-learn.

## C    BENCHMARK RESULTS ON ALL DATASETS

| Method | adult | census | loan | cabs | kings | news |
|---|---|---|---|---|---|---|
| | F1 | F1 | Macro | Macro | $R^2$ | $R^2$ |
| Identity | 0.677 | 0.648 | 0.664 | 0.756 | 0.647 | 0.038 |
| MedGAN | 0.045 | 0.000 | 0.548 | 0.328 | -4.758 | -1.182 |
| TableGAN | 0.496 | 0.367 | 0.511 | 0.340 | 0.547 | -0.521 |
| CTGAN | 0.628 | 0.459 | 0.637 | 0.576 | -0.761 | 0.006 |
| TVAE | 0.611 | 0.464 | 0.380 | 0.385 | 0.531 | -0.055 |
| Causal-TGAN | **0.662** | **0.509** | **0.661** | **0.684** | **0.563** | **0.025** |

Table 5: Machine learning efficacy on real-world datasets.

| Method | asia | | alarm | | child | | insurance | | ecoli70 | | arth150 | | healthcare | | mehra | |
|---|---|---|---|---|---|---|---|---|---|---|---|---|---|---|---|---|
| | KLD | LC | KLD | LC | KLD | LC | KLD | LC | KLD | LC | KLD | LC | KLD | LC | KLD | LC |
| MedGAN | 1.806 | -2.108 | **-0.081** | -2.125 | **-0.068** | -2.563 | **-0.033** | -2.449 | 7.088 | -1.396 | N/A | -1.417 | 1.067 | -1.573 | 0.734 | -1.996 |
| TableGAN | 0.635 | -2.612 | 0.237 | -3.935 | 0.178 | -3.651 | 0.069 | -4.34 | **0.243** | **-6.888** | 1.754 | -6.114 | 0.256 | -2.841 | 0.7 | -1.94 |
| TVAE | 0.054 | -3.12 | 0.168 | -4.537 | 0.108 | -4.999 | 0.088 | -4.446 | 1.084 | -5.597 | 2.187 | -2.095 | 0.042 | -3.587 | 0.154 | **-2.08** |
| CTGAN | 0.344 | **-3.668** | 0.489 | -2.631 | 0.141 | -3.126 | 0.091 | -3.501 | 1.613 | -2.639 | 1.538 | -2.819 | 0.232 | -3.682 | 0.478 | -2.035 |
| Causal-TGAN | **0.012** | -3.454 | 0.084 | **-5.381** | 0.061 | **-6.689** | 0.06 | **-6.472** | 0.502 | -6.847 | **0.28** | **-6.382** | **0.007** | **-3.817** | **0.11** | -2.046 |

Table 6: Benchmark results on simulated datasets.

