# OpenReview forum: "Causal-TGAN: Modeling Tabular Data Using Causally-Aware GAN"
_ICLR.cc/2022/Workshop/DGM4HSD — ICLR 2022 DGM4HSD workshop Poster_

### Official Review · Reviewer_A4Zd · 2022-03-24
**Good improvement over baseline but poorly described model**

**Rating:** 6
**Confidence:** 3

**Review:**

**Goals**: The goal of this work is to generate tabular data that incorporates inter-variable causal relationships as defined by a domain expert. The learning task is to train a neural network that functions as an autoregressive generator to model a structured causal model. In the case of partial knowledge, the goal is to train a conditional GAN that is conditioned on the observed data with causal relationships.

**Description**: The description is not clear. It is not stated what kind of neural network is being used for the autoregressive generator, G (equation 1). The architecture for the conditional GAN is also not described. In equation 2, it is unclear what C_cond is. The authors make no mention of how adversarial training was performed and do not even define the discriminator or its loss. The description of mode specific normalization in Section 4.1 is hard to follow and appears to not be novel as it has already been applied to a very similar problem.

**Evaluation**: The authors evaluate their work on real and simulated datasets. They compare performance to baselines in the literature. The authors define ML efficiency, KL divergence and Log-cluster as metrics to use for evaluation. The ablation study to compare relative utility of each of the different model components was useful in understanding the improvements seen over benchmark performance.

**Significance**: The problem they identify is a relevant one and it would be very useful to be able to incorporate expert knowledge into tabular data generation through the use of a causal graph. However, as mentioned above many of the technical details are unclear and could be better expanded upon, which makes it difficult to validate their approach. Having said that, assuming the technical details are sound, the improvements seen over baselines is significant and thus this model would be a valuable contribution to the field.

**Clarity**:The paper is well structured but the language is not clear to follow in many sections. The authors also rely heavily on knowledge of existing work in the field and do not attempt to elaborate on existing methods, even though these methods form an integral part of their own approach. There are also several spelling and grammatical errors.

---

### Official Review · Reviewer_tA8m · 2022-03-24
**Limited novelty and quesionable empirical performance**

**Rating:** 5
**Confidence:** 3

**Review:**

In this paper, the authors proposed Casual-TGAN, a generative model to generate tabular data given casual relationships provided by experts.

Pro:
- A novel contribution of this paper is the conditional generation given part of the information, which makes sense to me. This would be useful for completing missing data.
- The empirical results in simulated and real-world data are advantageous.
- The article is well-written.

Cons:
- The model closely follows Casual-GAN and has limited novelty in the generative model itself.
- Though the authors emphasized their model is specialized for tabular data, the narrative of the paper cannot persuade me that the model makes good use of the structure of tabular data. The model seems general and can be applied to any vectorized data without modification.
- The metric for real-world data is not convincing. In generative modeling, we should avoid reproducing identical data. But the metrics such as MLe or KLD would seem good even the generated data are identical to the training data. Simply having superior results on those metrics is not convincing.
- The casual reaaltions are required as input to construct the SCM, however, we don't always have that knowledge practically. This limits the practical application of this model.

Overall, I think this paper proposed a generative model with marginal improvement from previous work. Though the authors showed a superior empirical performance, the metrics used are questionable in my opinion. Thus I tend to suggest rejecting this paper.

---

### Official Review · Reviewer_9JP8 · 2022-03-27
**CAUSAL-TGAN: MODELING TABULAR DATA USING CAUSALLY-AWARE GAN**

**Rating:** 6
**Confidence:** 5

**Review:**

In this paper, the authors propose a causally-aware generator architecture, Causal-TGAN, to capture the relationships among variables (continuous-, discrete-, and mixed-type) by explicitly modeling pre-defined inter-variable causal relationships. Judea Pearl's Structural Causal Model (SCM) is utilized as the conceptual framework for the two main components: fitting separate neural networks Gi to capture each causal mechanism fi, and then generating samples autoregressively per the topological order. In the case of incomplete knowledge on the relationships among the variables, a conditional GAN is utilized to generate the rest of the variables conditioned on the variables used by Causal-TGAN.

Experiments are conducted on both synthetic and real datasets, and in comparison with existing models. The experiments include an ablation study. The experiments are rigorous and suggest that Causal-TGAN represents an improvement over other methods.

Overall, this is a technically-rigorous submission, but here are some concerns that somewhat diminish my enthusiasm about the novelty and impact of the proposed model:
1. The introduction never formally defines the problem. It is all introduced rather colloquially.
2. The paper also does not explicitly frame the scope. There are many complex models of causality. Vol-31, "Correlation and Causation in Biological Systems with Applications to Asymmetry" shows some biological models of causality, which include "backward" edges. The authors need to properly define the scope.
3. Are there some assumptions on the observed relationships among the variables? Are these presumed to be linear?
4.  What motivates this architecture, in light of the existing models? What is the deep insight over related work? The authors seem to indicate that it is mainly the variable encoding? Is that what drives the improved performance? It seems to be a very superficial argument.
5. I do not see a link to the source code for reproducibility.

---

### Decision · Program_Chairs · 2022-03-28

Accept (Poster)